# Effect of opioid-free anesthesia combined with pectoral nerve block on the quality of recovery in patients after mastectomy: A randomized, controlled trial

**Jiawei Chen[1]◎, Lewei He[2]◎, Yuying Shi[1], Jing Jiao[1], Shaoqiang Huang[1], Jianhua Zhou[3]‡\*, Qingyan Luo◎[1]‡\***

**1** Department of Anesthesiology, Obstetrics and Gynecology Hospital of Fudan University, Shanghai, China, **2** Department of Gynecology, Obstetrics and Gynecology Hospital of Fudan University, Shanghai, China, **3** Department of Anesthesiology, Huashan Hospital Fudan University, Shanghai, China

◎ These authors contributed equally to this work.
‡ JZ and QL also contributed equally to this work.
\* wsqlqy@126.com

## Abstract

### Objective

To evaluate the impact of opioid-free anesthesia (OFA) combined with regional blocks on the quality of recovery (QoR) in patients who underwent mastectomy.

### Methods

This randomized controlled trial involved 132 mastectomy patients who were randomized to receive either OFA combined with PECS block or opioid-based anesthesia (OBA) combined with PECS block. The QoR was assessed using the QoR-15 global score at 24 h post-surgery. Secondary outcomes included postoperative sufentanil consumption, incidence of postoperative nausea and vomiting (PONV), Numerical Rating Scale (NRS) scores at 1, 4, and 24 h, incidence of postoperative adverse events, extubation, incidence of severe bradycardia and intraoperative mean arterial pressure (MAP) and heart rate (HR) at after entering the operating room (T0, baseline value), after intubation (T1), after skin incision (T2), and after extubation (T3).

### Results

The QoR-15 global score at 24 h was not significantly different between groups (MD = -0.4, 95% CI [-3.8 to 4.7], $P = 0.67$). Postoperative sufentanil consumptions ($P = 0.075$), the incidence of PONV ($P = 0.12$), NRS scores at 1 h ($P = 0.36$), 4 h ($P = 0.53$), and 24 h ($P = 0.02$) were not significantly different. Incidence of adverse events (OR = 0, 95% CI [0 to 0.44], $P = 0.0063$) were lower in Group OFA than that in Group OBA. Extubation time was significantly longer in Group OFA than in Group OBA (MD = 15, 95%CI [10–18], $P < 0.001$). MAPs at T1 and T2 were significantly higher in Group OFA than in Group OBA ($P < 0.0125$), while MAP and HR at T3 were significantly lower in Group OFA than in

**Data availability statement:** All relevant data are within the paper and its Supporting information files.

**Funding:** The author(s) received no specific funding for this work.

**Competing interests:** The authors have declared that no competing interests exist.

Group OBA($P$ < 0.0125). Incidence of severe bradycardia were not significantly different ($P$ = 0.67).

## Conclusion

In conclusion, while OFA contributes to a reduction in adverse events, its integration with PECS blocks does not improve QoR or postoperative analgesia at 24 h post-mastectomy. Moreover, OFA was associated with a prolonged extubation time.

## Trial registration

chictr.org; registration number: ChiCTR2100043575.

## Introduction

Pain and postoperative nausea and vomiting (PONV) are significant barriers to the recovery of patients undergoing mastectomy [1]. The use of regional blocks has been shown to reduce postoperative pain [2], minimize opioid use, decrease the occurrence of PONV, and enhance overall recovery trajectories [3].

Opioids play a critical role in the perioperative period but are not without their drawbacks. They are linked to a range of adverse effects, including respiratory depression, PONV, and hindered patient recovery [4]. In light of these concerns, opioid-free anesthesia (OFA), when combined with a variety of antinociceptive agents and/or regional blocks regional blocks that eschew intraoperative opioid administration, may expedite the recovery process [5]. OFA has been shown to reduce opioid-related adverse events and enhance postoperative pain management, leading to a significant improvement in the quality of recovery (QoR) across different surgical procedures [6–8]. In the context of mastectomy, OFA has been correlated with diminished postoperative pain and elevated patient-reported QoR-40 scores at the 24-hour mark post-surgery [9]. However, the study in question did not incorporate regional blocks, which are recognized as highly effective in managing postoperative pain [10] and are especially prevalent in mastectomy cases.

The integration of regional blocks is considered a cornerstone of contemporary multimodal analgesia strategies for breast surgery, including mastectomy. The rationale for combining regional blocks with OFA is to further enhance analgesia and potentially reduce opioid-related adverse events, aligning with current recommendations that advocate for a comprehensive approach to pain management. The potential benefits of incorporating OFA into regional blocks warrant further investigation. However, few publications have evaluated the effectiveness of the OFA in combination with regional block during mastectomy.

Hence, the primary hypothesis in this research is that OFA combined with regional block will improve the global score of the patient-reported QoR-15 questionnaire measured at 24 h postoperatively compared to opioid-based anesthesia (OBA). The NRS score, cumulative opioid consumption, incidence of PONV, and incidence of other adverse side effects within 24 h were also studied as secondary outcomes.

## Methods

Ethical approval for this study (2020–153) was provided by the Ethical Committee of the Obstetrics and Gynecology Hospital of Fudan University. This prospective, randomized single-center controlled study was registered before patient enrollment at chictr.org (study ID: ChiCTR2100043575, 22/02/21). The trial was performed at the Obstetrics and Gynecology

Hospital of Fudan University, Shanghai, China, from March 7th, 2021 to January 1st, 2022. Written informed consent was obtained from all subjects. And participating patients could withdraw from the trial at any time. All procedures were carried out in accordance with relevant guidelines, regulations, and CONSORT recommendations.

## Patients

This study included patients aged 18–70 years with an American Society of Anesthesiologists (ASA) physical status of I-II undergoing total or partial mastectomy, with axillary lymph node dissection or sentinel lymph node biopsy, which typically requires general anesthesia.

Exclusion criteria included: a. Patients with impaired kidney or liver function; b. body mass index (BMI) > 35 kg/m$^2$; c. allergy to the study medication; d. pregnancy or lactation; e. chronic cardiovascular disease; f. chronic pain; g. history of abnormal anesthesia; h. preoperative use of opioids or psychotropic drugs; i. inability to understand the numerical rating scale (NRS); j. inability to use patient-controlled analgesia (PCA); k. refusal to participate.

## Allocation and blinding

The participants were randomly allocated to either Group OFA or Group OBA by means of a draw of concealed envelopes detailing group designation. The participants were provided with either the OFA protocol or OBA protocol according to the allocation, and all the participants received the PECS block before surgery. The randomization was performed at a 1: 1 ratio using the PASS 11.0. When patients arrived in the OR, an anesthesiologist unsealed the envelopes to present the treatment allocation. Then, the same anesthesiologist prepared trial the drugs and took charge of the intraoperative courses. There was no masking of the groups for either the anesthesiologist or the PACU nurses. To minimize bias, they did not participant in assessing patients during the study. Surgeons, patients and all involved in the surgical ward, including physicians and nurses, and investigators who conducted the outcome assessment, had no knowledge of group allocation.

## Anesthesia protocol

Once the patients were moved to the operating room, regular monitoring comprising ECG, SpO$_2$, and noninvasive blood pressure (BP), was implemented. Spectral entropy including state entropy (SE) was also applied.

After that, ultrasound-guided pectoral nerve (PECS) blocks were performed by an anesthesiologist blinded to the group allocation. This anesthesiologist injected 10 ml of 0.4% ropivacaine between the pectoralis muscle and 20 ml of 0.4% ropivacaine above the serratus anterior muscle for the PECS block, as previously described [11]. After 30 minutes observation, the sensory level of the block was assessed with a cold swab in each dermatomal distribution from T2 to T6. If the cold sensation in the segment did not decrease or disappear, then the block was considered unsuccessful [12] and the patient withdrew from the study.

In Group OFA, the patients were given intravenous 1 μg/kg of dexmedetomidine over 10 min. Meanwhile, the patients were given intravenous 0.5 μg/kg of sufentanil in Group OBA. Anesthesia was induced with propofol (target-controlled infusion, TCI, Marsh mode, six μg/ml) and 0.6 mg/kg of rocuronium. An endotracheal tube was inserted thereafter. Forty milligrams of parecoxib were given after induction. Continuous intravenous propofol infusion using TCI at an effect site of 2–4 μg/ml was administered to sustain a goal SE value between 40–60 for maintaining anesthesia. Moreover, to maintain the MAP and HR within 20% of the baseline level, 0.5–1.5 μg/kg/h of dexmedetomidine in Group OFA or repeated 0.025 μg/kg of sufentanil in Group OBA was administered. In situations where the titration could not obtain

the goal MAP and HR, vasoactive agents were used as determined by the anesthesiologist for intraoperative anesthesia. Mechanical ventilation was managed with a ventilator, and the respiratory parameters were modified to retain an end-tidal carbon dioxide between 35 and 45 mmHg. Muscle relaxation was sustained by administering fifteen milligrams of rocuronium every 40 minutes until the surgery was complete. Although neuromuscular monitoring was not employed in this study due to the unavailability of monitoring equipment, the anesthesiologist relied on clinical judgment and standard intraoperative assessments to ensure appropriate muscle relaxation throughout the procedure.

Once the surgery was complete, all anesthetics were ceased. Neostigmine (0.02 mg/kg) and atropine (0.01 mg/kg) were administered. All patients were transferred to the PACU after extubation.

All patients received the same standardized postoperative multimodal analgesia regimen in our institution. In detail, 40 mg parecoxib was given every 12 h until discharge from the hospital. PCA pumps were administered with a 2 µg of sufentanil bolus at maximum demand every 10 minutes without continuous infusion until 24 h after surgery. Considering the increasing trend towards ambulatory breast surgery, our institution has re-evaluated its postoperative analgesic regimen. While PCA pumps were utilized in this study, we acknowledge that oral opioid alternatives may offer greater clinical relevance in the context of same-day discharge protocols now prevalent in many centers. Before surgery, every patient was required to use the PCA pump and report their pain intensity utilizing the NRS, with 0 indicating "no pain" and 10 indicating "worst imaginable pain." After being moved to the PACU, a PCA pump was connected to the patient. Then, the patient was given permission to administer PCA.

Dexamethasone (5 mg) and palonosetron (0.075 mg) were given to all patients to prevent PONV. PONV was managed by physicians in the ward.

## Outcomes

The primary outcome was the global score of the Chinese version QoR-15 [13] measured at 24 h postoperatively.

The QoR-15, a comprehensive patient-reported tool employed to assess functional recovery, comprises fifteen questions assessing five domains of recovery, including pain, physical comfort, physical independence, psychological support, and emotional state. The global QoR-15 score ranged from 0 (poor) to 150 (excellent). A score greater than or equal to 118 was considered consistent with a good recovery [14].

The QoR-15 was administered on the day before surgery (baseline) and at 24 h. The patients answered the QoR questionnaire in the surgical wards instructed by an investigator blinded to the allocation of groups.

The secondary outcomes were postoperative pain scores, postoperative sufentanil consumption, postoperative nausea and vomiting (PONV), and other adverse events within the first 24 h, which included but were not limited to respiratory depression, dizziness and pruritus. Postoperative pain scores were measured using the NRS at rest at 1, 4, and 24 h after surgery. Data such as extubation time, severe bradycardia (HR less than 45 bpm) requiring the administration of atropine, intraoperative MAP and HR at after entering the operating room (T0, baseline value), after intubation (T1), after skin incision (T2), and after extubation (T3) were also recorded. Patient demographic data; estimated blood loss; operation duration; and intraoperative propofol, dexmedetomidine, and sufentanil were also recorded.

## Statistical analysis

Our power calculation was predicated on the QoR-15 score at 24 h postoperatively, using data from Bu et al.'s study [13], which validated the QoR-15 instrument in a surgical population.

We selected this control value for our power calculation to ensure our study was adequately powered to detect a clinically meaningful difference in recovery quality. According to Bu et al.'s study, the QoR-15 score at 24 h after general anesthesia was 106.05 ± 15.73. Furthermore, eight is considered to be the MCID [15]. The sample size was calculated to detect at least eight differences in the QoR-15 score with an alpha level of 0.05 and 80% power. A sample size of 63 patients per group was needed The sample size was expanded to 136 patients to compensate for any dropouts.

Statistics were performed with IBM SPSS 29.0 and GraphPad Prism 7.0 for Windows. An absolute standardized difference (ASD) was used to compare baseline characteristics. An ASD greater than 0.34 was considered imbalanced, as calculated by $1.96 \times \sqrt{\dfrac{n_1 + n_2}{n_1 \times n_2}}$ [16]. The normality of continuous variables was evaluated using the Shapiro-Wilk test. Continuous variables are presented as the means ± SDs or medians with IQRs depending on whether they were normally distributed. Student's t-test was used for analysis of normally distributed variables and Mann–Whitney U-test was used for analysis of nonnormally distributed variables. QoR-15 score was not normally distributed but was reported using means ± SD because the value of MICD expressed as the means ± SD [15]. The variation between the global QoR-15 scores before the surgery and after was reported as the global QoR-15 scores at 24 h after the surgery minus that before the surgery. Repeated measures analysis of variance (ANOVA) was performed to compare variables at multiple time points between groups. Categorical variables are presented as numbers (%). $\chi^2$ tests were used to compare categorical variables. 95% CIs were estimated using the Hodges–Lehmann estimate. The median difference (MD) results are reported as Group OFA minus Group OBA. All *P* values were two-sided. A *P* value less than 0.05 was considered statistically significant. Bonferroni correction was used for analysis of the NRS scores as well as intraoperative MAP and HR. Statistical significance was adjusted to *P* < 0.017 for the NRS scores, *P* < 0.0125 for intraoperative MAP and HR.

## Results

From 142 eligible patients, 136 patients were consented and randomized to groups of 68 each. After randomization, three patients in Group OFA were excluded. One patient in Group OFA was excluded due to withdrawal of approval before surgery. PECS block was successful in all patients. Two patients in Group OFA had their surgeries canceled and were excluded because they had fever on the day of surgery. One patient in Group OBA did not complete QoR-15 questionnaire at 24 h, but the other outcomes were complete. Thus, the analysis of primary outcome included 132 patients (Fig 1).

### Demographic data and surgical data

The baseline demographic data and surgical data were not significantly different between the groups, see Table 1.

The propofol doses in Group OFA was significantly greater than that in Group OBA (*P* = 0.0016). Other intraoperative data were not significantly different between the groups, see Table 2.

### QoR-15 score at 24 h

The baseline QoR-15 global score was not significantly different between groups (ASD = 0.073, see Table 1). The 24-hour QoR-15 global score was not significantly different (*P* = 0.46). The different dimensions of the QoR-15 as well as the variation between the QoR-15 before the surgery and after were also not significantly different between groups, see Table 3. There were

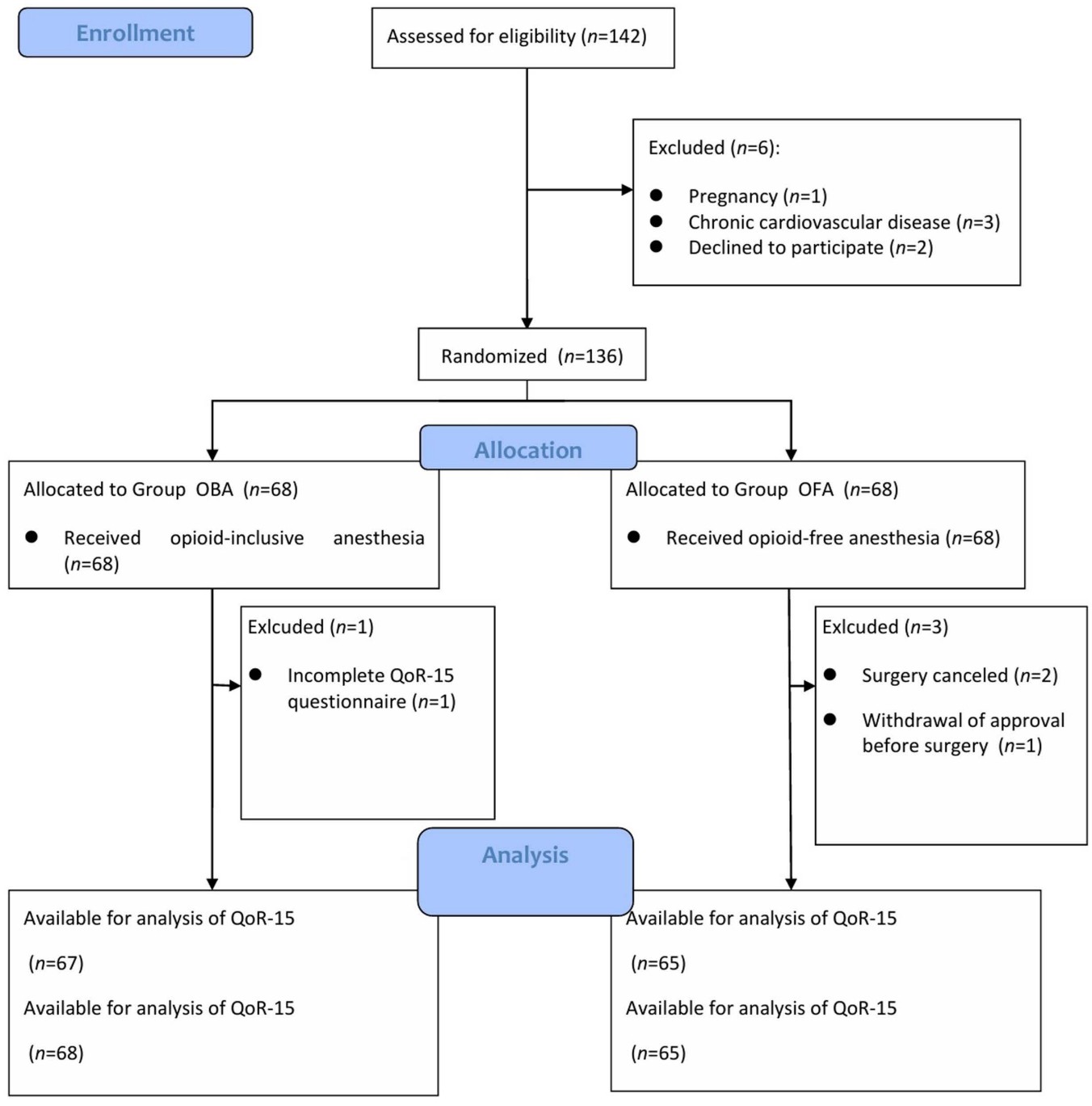

**Fig 1. CONSORT flow diagram.**

54 patients with a QoR-15 global score ≥ 118 at 24 h in Group OFA and 57 in Group OBA (RR = 0.98, 95% CI [0.83 to 1.1], $P = 0.75$).

### Postoperative analgesia and adverse events

Postoperative sufentanil consumption within 24 h was not significantly different between the groups ($P = 0.075$). The NRS scores were not significantly different between groups after the

**Table 1. Demographic and surgical characteristics.**

| Variable | Group OFA (*n* = 65) | Group OBA (*n* = 68) | ASD |
|---|---|---|---|
| **Age (y)** | 46.4 ± 7.5 | 48.0 ± 8.6 | 0.20 |
| **Weight (kg)** | 60.0 ± 6.7 | 60.3 ± 6.7 | 0.045 |
| **Height (cm)** | 161.7 ± 5.9 | 160.9 ± 4.4 | 0.15 |
| **BMI (kg/m²)** | 23.1 ± 2.3 | 23.6 ± 3.3 | 0.18 |
| **Procedure** | | | |
| **Mastectomy with SLNB** | 30 (46.2) | 34 (52.3) | 0.12 |
| **Mastectomy with ALND** | 17 (26.2) | 14 (21.5) | 0.11 |
| **Partial mastectomy with SLNB** | 15 (23.1) | 13 (20) | 0.075 |
| **Partial mastectomy with ALND** | 3 (4.6) | 4 (6.2) | 0.071 |
| **Pre-surgery Global QoR-15 scores** | 140.3 ± 8.0 | 139.6 ± 10.8 | 0.073 |

ALND = axillary lymph node dissection; ASD = absolute standardized difference; BMI = body mass index; SLNB = sentinel lymph node biopsy.

Data are shown as mean±SD, or proportion (*n*[%]).

ASD larger than 0.34 was considered imbalanced, as calculated by $1.96 \times \sqrt{\dfrac{n_1 + n_2}{n_1 \times n_2}}$ [16], where $n_1$ and $n_2$ were per-group sample sizes.

**Table 2. Intra-operative data.**

| Variable | Group OFA (*n* = 65) | Group OBA (*n* = 68) | P value |
|---|---|---|---|
| **Operation duration (min)** | 120 (100–140) | 110 (90–125) | 0.075 |
| **Estimated blood loss (mL)** | 30 (20–50) | 30 (20–30) | 0.546 |
| **Propofol dose (mg)** | 1040 (900–1190) | 890 (677.5 to 1070) | 0.0016 |
| **Sufentanil dose (μg)** | – | 30 (27–40) | – |
| **Dexmedetomidine dose (μg)** | 90 (74 to 107.5) | – | – |

ASD = absolute standardised difference; QoR = quality of recovery.

Data are shown as mean±SD, median (IQR).

**Table 3. The 15-item quality of recovery questionnaire global and dimensions scores at 24 h.**

| Variable | Group OFA (*n* = 65) | Group OBA (*n* = 67) | Mean difference (95% CI) | P value |
|---|---|---|---|---|
| **Global QoR-15 scores** | 126.9 ± 10.3 | 127.4 ± 14.0 | -0.4 (-3.8 to 4.7) | 0.67 |
| **Pain** | 19.8 ± 0.8 | 19.1 ± 2.7 | 0.7 (-1.4 to 0.01) | 0.053 |
| **Physical comfort** | 42.5 ± 4.3 | 43.4 ± 5.5 | -0.9 (-0.8 to 2.6) | 0.29 |
| **Physical independence** | 8.8 ± 5.8 | 9.1 ± 5.0 | -0.3 (-1.5 to 2.2) | 0.74 |
| **Psychological support** | 19.8 ± 1.2 | 19.7 ± 1.5 | 0.2 (-0.7 to 0.3) | 0.47 |
| **Emotional status** | 36.0 ± 4.2 | 36.1 ± 6.1 | -0.07 (-1.7 to 1.8) | 0.94 |
| **Variation between the global scores before the surgery and after** | -13.4 ± 10.1 | -12.2 ± 12.2 | -1.2 (-5.1 to 2.7) | 0.54 |

QoR = quality of recovery.

Data are shown as mean±SD.

surgery ($P > 0.02$), see Table 4. According to repeated measures ANOVA, there was not enough evidence to show a statistically significant interaction effect between group allocation and time for NRS score ($P = 0.139$). There was a significant effect of group on the NRS score (F = 3.5, $P = 0.033$). There was not enough evidence of a significant main effect of time on the NRS score (F = 0.8, $P = 0.45$).

Four patients in Group OBA reported PONV, while no patient in Group OFA reported PONV ($P = 0.12$). Three patients in Group OBA reported dizziness and one patient reported pruritus. Moreover, no adverse events were reported in Group OFA. No other adverse events were observed. The incidence of postoperative adverse events was significantly lower in Group OFA than in Group OBA ($P = 0.0062$), see Table 4.

## Additional interests

Extubation time was significantly longer in Group OFA than in Group OBA ($P < 0.001$). In both groups, the means of intraoperative MAP and HR were inside the range from 50 to 100, mmHg and bpm. MAPs at T1 and T2 were significantly higher in Group OFA than in Group OBA ($P < 0.0125$), while MAP and HR at T3 were significantly lower in Group OFA than in Group OBA ($P < 0.0125$), see Table 5. According to repeated measures ANOVA, there was a statistically significant interaction effect between group allocation and time for MAP ($P$

**Table 4. Post-operative outcomes within 24 h.**

| Variable | Group OFA ($n = 65$) | Group OBA ($n = 68$) | Median difference (95% CI)/ OR (95% CI) | P value |
|---|---|---|---|---|
| **Post-operative sufentanil consumption (μg)** | 0 (0–2) | 2 (0–2) | 0 (-1.4 to 0) | 0.075 |
| **Pain scores** | | | | |
| **At 1 h** | 0 (0–0) | 0 (0–0) | 0 (0–0) | 0.36 |
| **At 4 h** | 0 (0–0) | 0 (0–0) | 0 (0–0) | 0.53 |
| **At 24 h** | 0 (0–0) | 0 (0–1) | 0 (0–0) | 0.02 |
| **Incidence of PONV** | 0 (0) | 4 (5.9) | 0 (0 to 0.75) | 0.12 |
| **Incidence of post-operative adverse events** | 0 (0) | 8 (11.8) | 0 (0 to 0.44) | 0.0063[a] |

PONV=post-operative nausea and vomiting, LOS=length of stay.

Data are shown as median (IQR) or proportion ($n$[%]).

$P < 0.017$ for NRS scores.

[a]: Statistically significant difference between groups.

**Table 5. Additional interest.**

| Variable | Group OFA ($n = 65$) | Group OBA ($n = 68$) | Mean/median difference (95% CI)/ OR (95% CI) | P value |
|---|---|---|---|---|
| **Extubation time (min)** | 10 (5 to 13.8) | 25 (15–30) | 15 (10–18) | <0.001 [a] |
| **Intraoperative MAP** | | | | |
| At T0 | 94.5 ± 10.9 | 91.9 ± 13.1 | 2.6 (-1.5 to 6.7) | 0.214 |
| At T1 | 80.8 ± 13.9 | 93.6 ± 13.2 | -12.9 (-17.6 to -8.2) | <0.001 [a] |
| At T2 | 78.9 ± 12.5 | 91.6 ± 13.2 | -12.6 (-16.7 to -8.6) | <0.001 [a] |
| At T3 | 95.9 ± 13.4 | 89.0 ± 11.0 | 6.8 (2.6 to 11.0) | 0.002 [a] |
| **Intraoperative HR** | | | | |
| At T0 | 75.4 ± 12.8 | 70.7 ± 10.9 | 4.7 (0.6 to 8.7) | 0.025 |
| At T1 | 70.6 ± 12.6 | 73.4 ± 10.3 | -2.8 (-6.7 to 1.2) | 0.171 |
| At T2 | 65.6 ± 11.3 | 69.5 ± 9.5 | -3.9 (-7.5 to -0.3) | 0.33 |
| At T3 | 76.2 ± 11.2 | 65.8 ± 9.4 | 10.4 (6.8 to 14.0) | <0.001 [a] |
| **Severe bradycardia** | 3 [4.5] | 4 [6.2] | 1.4 (0.36 to 5.3) | 0.67 |

MAP=mean arterial pressure, HR=heart rate, T0 = after entering the operating room, T1 = after intubation, T2 = after skin incision, T3 = after extubation.

Data are shown as mean±SD, median (IQR) or proportion ($n$[%]).

$P < 0.0125$ for intraoperative MAP and HR.

[a]: Statistically significant difference between groups.

< 0.001) and HR ($P < 0.001$). There was a significant main effect of group allocation and time on MAP ($P < 0.001$ for group allocation, $P < 0.001$ for time) and HR ($P < 0.001$ for group allocation, $P < 0.001$ for time).

Seven patients had severe bradycardia (HR less than 45 bpm) and recovered after 0.25–0.5 mg of atropine. No other intraoperative adverse events were reported.

## Discussion

To our knowledge, this is the first published study to evaluate the effects of OFA combined with regional block on the QoR after mastectomy. Effect of OFA on cumulative opioid consumption, pain scores, and opioid-related adverse events were also evaluated. Contrary to our expectations, OFA didn't improve the QoR-15 score at 24 h compared with OBA, although OFA reduced adverse events. OFA had no effect on the cumulative opioid consumption, NRS scores, or incidence of PONV after surgery, but it was correlated with a prolonged extubation time.

The primary advantage of OFA lies in its ability to avoid the use of opioids, thereby mitigating associated complications such as respiratory depression, gastrointestinal disturbances, and the potential for addiction or dependence [17]. However, these benefits may be less pronounced when other opioid-sparing strategies, such as regional blocks, are effectively employed. This is consistent with findings from Chassery et al. [18] and Mieszczański et al. [19], who observed limited advantages of OFA in the context of multimodal analgesia. Among these strategies, the PECS block is a cornerstone for breast surgery, providing effective pain relief and reducing the incidence of PONV [20]. In our study, the combination of PECS blocks with minimal sufentanil use achieved excellent analgesia and a low rate of adverse events. However, the efficacy of PECS blocks may overshadow any additional benefits of OFA on QoR, despite prior studies suggesting that OFA could improve QoR [17]. Notably, Ibrahim et al. [7] found that while OFA combined with regional blocks enhanced QoR in the short term, these benefits were transient and diminished after six hours. Our findings suggest that when high-quality regional anesthesia, such as PECS blocks, is effectively utilized, the added benefits of OFA on QoR may be marginal and statistically negligible. This indicates that the routine use of OFA may not be necessary in such scenarios.

The results of the ANOVA revealed significant differences between the two groups with respect the NRS scores ($P = 0.033$). Intriguingly, we observed a difference trend towards lower NRS scores at the 24 h in the OFA group, yet this decrease did not achieve statistical significance ($P = 0.02$). Additionally, the NRS scores at 1 and 4 h post-surgery, as well as the total sufentanil consumption, demonstrated no significant differences between the groups, with $P > 0.36$. While OFA may indeed offer some enhancement in postoperative analgesic effects, the magnitude of this improvement may be too slight to be discerned within our study's sample size. Moreover, the similarity in the QoR-15 pain dimension scores at the 24-hour postoperative period between groups suggests that any difference in this regard is likely of little clinical consequence.

Due to the minimal overall use of perioperative sufentanil coupled with effective dual prophylaxis against PONV, only a small number of patients, four in total, experiencing this condition. As a result, we did not observe a statistically significant difference in the incidence of PONV ($P = 0.12$), despite previous reports indicating that OFA significantly reduces PONV [17,21]. Consequently, our findings do not clearly demonstrate the anticipated advantage of OFA in mitigating PONV. However, when evaluating the overall incidence of adverse events, our study revealed a significantly lower rate in the OFA group ($P = 0.0062$). Notably, these adverse events were minor and did not hinder patient recovery. This is evidenced by the comparable QoR-15 scores for physical comfort observed between the groups, suggesting that

despite the lower incidence of adverse events with OFA, the overall impact on recovery was not substantial.

Despite the potential benefits of OFA, intraoperative adverse events can influence its clinical adoption. Studies have indicated that OFA protocols involving dexmedetomidine are sometimes linked to prolonged extubation times and increased incidences of bradycardia [22, 23]. In our trial, the OFA group experienced significantly longer extubation times compared with the OBA group ($P < 0.001$), which is consistent with the reported challenges of OFA.

The premature termination of the POFA study [23], due to severe bradycardia in the OFA group, underscores the importance of carefully considering the dosing and administration of anesthetic agents. In our study, we deliberately avoided high doses of dexmedetomidine, largely due to the robust analgesic properties provided by PECS block. The total dexmedetomidine dosage we utilized was a modest 90 μg (74 to 107.5 μg), which equates to an infusion rate of $0.78 \pm 0.3$ μg/kg/h. This dosage is significantly lower than the $1.2 \pm 0.2$ μg/kg/h reported in the POFA study. In our study, only 7 patients suffered severe bradycardia (HR less than 45 bpm) and recovered after 0.25–0.5 mg of atropine. As a result, we did not find a difference in intraoperative adverse events between the groups.

There are several limitations to our study. First, the anesthesiologist in charge of administering anesthesia was not blinded. Because of the differences in the effects of the drugs, it's impossible for an anesthesiologist to not identify the groups. To avoid biases, all patients, physicians, nurses in the ward, surgeons, and investigators assessing outcomes were blinded. Second, the results may be limited to patients using the OFA protocol with dexmedetomidine alone. In our study, the OFA protocol included intravenous dexmedetomidine and PECS. At present, the optimal OFA protocol remains controversial. Moreover, in a pilot study, we found that this OFA regimen was sufficient for surgery. Thus, we did not apply other reported agents. The combination of these different agents might have an additional effect on the results. Additionally, our postoperative analgesic strategy did not include the administration of acetaminophen, a widely recommended component in multimodal analgesia protocols that could potentially decrease opioid consumption. Despite its absence, we observed that the overall outcomes of our study, including pain scores and opioid-related adverse events, were not significantly affected. This suggests that while the inclusion of acetaminophen might have further optimized our analgesic regimen, its omission did not materially alter the principal findings related to the quality of recovery and the incidence of adverse events postoperatively. Future studies should incorporate such recommendations to enhance the generalizability of the results.

## Conclusions

In conclusion, while OFA contributes to a reduction in adverse events, its integration with PECS blocks does not improve QoR or postoperative analgesia at 24 h post-mastectomy. Moreover, OFA was associated with a prolonged extubation time. The benefits of OFA appear limited in the context of effective regional anesthesia, indicating that its use may be non-essential when high-quality regional blocks are administered.

## Supporting information

**S1 File. CONSORT checklist.**
(DOCX)

**S2 File. Accessible Data.**
(XLSX)

## Acknowledgment

We would like to express our sincerest appreciation for Yingxiu Pu, Hanli Hua, Minghui Xu, Juyun Zhu, Qinqwen Sun, Lin Tian, and Lingyun Fan in the Department of Anesthesiology of the Obstetrics and Gynecology Hospital of Fudan University. We also thank Kejing Wu, Shaomei Fu, Yuchun Jin, Hongliang Chen, Fuwen Wang, Ang Ding, Jian Sun, Peng Zhang and Yipeng Fu in the Department of Breast Surgery of the Obstetrics and Gynecology Hospital of Fudan University. Our gratitude also extends to all members of the 16th wards at Fudan University's Obstetrics and Gynecology Hospital.

## Author contributions

**Conceptualization:** Jiawei Chen, Lewei He, Jianhua Zhou, Qingyan Luo.

**Data curation:** Yuying Shi, Qingyan Luo.

**Formal analysis:** Jiawei Chen, Lewei He.

**Investigation:** Jiawei Chen, Yuying Shi, Jing Jiao, Shaoqiang Huang, Qingyan Luo.

**Methodology:** Yuying Shi, Qingyan Luo.

**Project administration:** Jianhua Zhou.

**Resources:** Jing Jiao, Shaoqiang Huang, Jianhua Zhou.

**Software:** Jiawei Chen.

**Supervision:** Jing Jiao, Shaoqiang Huang, Jianhua Zhou, Qingyan Luo.

**Validation:** Jiawei Chen, Lewei He, Yuying Shi, Jing Jiao, Shaoqiang Huang, Qingyan Luo.

**Visualization:** Jiawei Chen, Lewei He, Shaoqiang Huang, Qingyan Luo.

**Writing – original draft:** Jiawei Chen, Lewei He.

**Writing – review & editing:** Jiawei Chen, Lewei He, Jianhua Zhou, Qingyan Luo.

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
