## [Decision Letter · Decision Letter 0]

20 Dec 2024

PONE-D-24-44917Effect of opioid-free anesthesia combined with pectoral nerve block on the quality of recovery in patients after mastectomy: A randomized, controlled trialPLOS ONE

Dear Dr. Luo,

Thank you for submitting your manuscript to PLOS ONE. After careful consideration, we feel that it has merit but does not fully meet PLOS ONE’s publication criteria as it currently stands. Therefore, we invite you to submit a revised version of the manuscript that addresses the points raised during the review process.

We look forward to receiving your revised manuscript.

Kind regards,

Alessandro De Cassai, MD

Academic Editor

PLOS ONE

Journal Requirements:

 When submitting your revision, we need you to address these additional requirements. 1. Please ensure that your manuscript meets PLOS ONE's style requirements, including those for file naming. The PLOS ONE style templates can be found at https://journals.plos.org/plosone/s/file?id=wjVg/PLOSOne_formatting_sample_main_body.pdf and https://journals.plos.org/plosone/s/file?id=ba62/PLOSOne_formatting_sample_title_authors_affiliations.pdf. 2. We are unable to open your Supporting Information file [renamed_7e522.sav]. Please kindly revise as necessary and re-upload. 3. Please include captions for your Supporting Information files at the end of your manuscript, and update any in-text citations to match accordingly. Please see our Supporting Information guidelines for more information: http://journals.plos.org/plosone/s/supporting-information. 

Reviewers' comments:

Reviewer's Responses to Questions

**Comments to the Author**

1. Is the manuscript technically sound, and do the data support the conclusions?

Reviewer #1: Yes

Reviewer #2: Partly

2. Has the statistical analysis been performed appropriately and rigorously? 

Reviewer #1: Yes

Reviewer #2: Yes

3. Have the authors made all data underlying the findings in their manuscript fully available?

Reviewer #1: Yes

Reviewer #2: Yes

4. Is the manuscript presented in an intelligible fashion and written in standard English?

Reviewer #1: Yes

Reviewer #2: Yes

5. Review Comments to the Author

Reviewer #1: The paper was well written with respect to the statistical presentation.

The study was appropriately designed with power considerations requiring a sample size of 63 patients per group was and expanded to 136 patients to compensate for any dropouts. Both parametric and non parametric analysis approaches were performed as needed. Efficacy and adverse events were compared as per the two arms. The tables were well formatted and easily interpretable. Both groups were fairly well balanced. The analysis was simple with univariate comparisons.

As per the authors the groups were similar with respect to outcomes. That is, while OFA contributes to a reduction in adverse events, its integration with PECS blocks does not improve QoR or postoperative analgesia at 24 h post-mastectomy. Moreover, OFA was associated with a prolonged extubation time.

Reviewer #2: Dear Authors, thank you for the opportunity to review your study.

You have selected an interesting and contemporary topic that still needs a lot of detailed studies. I’ve also really appreciated the idea to incorporate regional anaesthesia, another current subject. However the manuscript presents important inaccuracies and requires some major revisions.

Firstly, the main outcome of the paper is not so innovative since ERAS protocols already promotes opioid alternatives to provide opioid reduced anesthesia (ORA) and OFA to patients.

Secondly, the reason why your main outcome refers to a study that doesn’t include mammary surgery is not clear (Bu eat al. “Validation of the Chinese version of the Quality of Recovery-15 Score and Its Comparison with the Post-Operative Quality Recovery Scale.”). Furthermore, in my opinion, the value of QoR-15 you used as baseline (106,5) is very low and does not represent the real quality of recovery of patients undergoing mastectomy beyond the use of opioids. This is also clear from your results: the lowest value of QoR-15 you have registered is 127.

In the end, I think it would’ve been interesting to analyse the variation between the QoR-15 before the surgery and after to evaluate if there were any differences between the two groups.

Obviously, changing the main outcome would change the sample size, can you clarify the choice you made?

In addition some minor revision that requires your attention:

- At page 5, lines 19-20; you should specify the type of surgery maybe writing “undergoing total or partial mastectomy, with axillary lymph node dissection or sentinel lymph node biopsy”;

- At page 6, lines 20-21; even if implicit, you should explain better the type of block you performed and the dosage and type of anaesthetic you used for each injection;

- At page 7, talking about muscle relaxation, include the neuromuscular monitoring;

- At page 8, clarify what you did include in “adverse events”;

- In table n.1 you should also consider the QoR-15 pre-surgery;

- Consider a rearrangement of the Discussion paragraph as it’s too long compared to the overall length of the article.

6. PLOS authors have the option to publish the peer review history of their article (what does this mean? ). If published, this will include your full peer review and any attached files.

**Do you want your identity to be public for this peer review?** For information about this choice, including consent withdrawal, please see our Privacy Policy .

Reviewer #1: No

Reviewer #2: **Yes: ** Marco Covotta

---

## [Author Response · Author response to Decision Letter 1]

11 Jan 2025

PLOS ONE Editorial Office

PLOS ONE

Dear Dr. De Cassai and the PLOS ONE Editorial Team,

Thank you for the opportunity to revise our manuscript titled "Effect of opioid-free anesthesia combined with pectoral nerve block on the quality of recovery in patients after mastectomy: A randomized, controlled trial" (Manuscript ID: PONE-D-24-44917). We appreciate the constructive feedback provided by you and the reviewers, which has helped us improve the quality of our work.

Below, we address each of the reviewers' comments and outline the changes made to the manuscript:

Reviewer #1 Comments:

We appreciate the reviewer's positive feedback regarding the statistical presentation and study design. We have ensured that all statistical analyses are clearly described in the revised manuscript.

Reviewer #2 Comments:

1.“Firstly, the main outcome of the paper is not so innovative since ERAS protocols already promotes opioid alternatives to provide opioid reduced anesthesia (ORA) and OFA to patients.”

- We acknowledge the concern regarding the innovation of our study. While ERAS protocols do promote opioid alternatives, our research specifically investigates the effects of opioid-free anesthesia (OFA) combined with regional blocks on the quality of recovery (QoR) after mastectomy. This focus on OFA in breast surgery is novel and provides specific data on its impact on postoperative recovery, pain management, and adverse events. Importantly, our study has already implemented several core measures of ERAS, including multimodal analgesia, PONV prevention, and total intravenous anesthesia (TIVA), which is recommanded by ERAS sociaty[1]. These elements not only enhance the QoR but also increase the relevance of our study as a reference for the implementation of ERAS protocols in clinical practice.

2.“Secondly, the reason why your main outcome refers to a study that doesn’t include mammary surgery is not clear (Bu eat al. “Validation of the Chinese version of the Quality of Recovery-15 Score and Its Comparison with the Post-Operative Quality Recovery Scale.”). Furthermore, in my opinion, the value of QoR-15 you used as baseline (106,5) is very low and does not represent the real quality of recovery of patients undergoing mastectomy beyond the use of opioids. This is also clear from your results: the lowest value of QoR-15 you have registered is 127.”

- We selected Bu et al.'s study[2] as a reference because it validated the Chinese version of the QoR-15 and was relevant to our patient population in mainland China. We clarify that Bu's study does include breast surgery, categorizing it under “General” surgery type in Table 1 in their study. Our findings of a higher postoperative QoR-15 score (127.4±14.0) compared to Bu's baseline score (106.5) reflect advancements in perioperative management, particularly in minimizing opioid use. We have recalculated the required sample size based on the QoR-15 scores from Zhang et al.'s study[3], which reported a postoperative 24-hour QoR-15 score of 132.0 ± 12.0 for patients receiving opioid-based anesthesia (OBA). Our sample size has been recalculated based on this new reference, ensuring that our study is adequately powered to detect clinically meaningful differences.

Sample Size Calculation:

Using the formula for comparing two means in clinical trials:

 and 

, .

with a MICD of 8, SD of 12, and a significance level of 0.05. Therefore, we propose a total sample size of 74 patients (37 per group) to detect a clinically meaningful difference in QoR-15 scores. This revised calculation ensures that our study is adequately powered to assess the impact of OFA combined with regional blocks on the quality of recovery in patients undergoing mastectomy.

3.“In the end, I think it would’ve been interesting to analyse the variation between the QoR-15 before the surgery and after to evaluate if there were any differences between the two groups.”

- We have added an analysis of the variation in QoR-15 scores before and after surgery in Table 3. The results show no statistically significant difference between the two groups, with changes of -13.4±10.1 in the OFA group versus -12.2±12.2 in the OBA group (MD=-1.2, P=0.54).

4.**Minor Revisions**:

We have made the following revisions as per your suggestions:

‘At page 5, lines 19-20; you should specify the type of surgery maybe writing “undergoing total or partial mastectomy, with axillary lymph node dissection or sentinel lymph node biopsy”;’

- Specified the types of surgeries performed (total or partial mastectomy, with axillary lymph node dissection or sentinel lymph node biopsy) on Page 5, Line 19-20.

“At page 6, lines 20-21; even if implicit, you should explain better the type of block you performed and the dosage and type of anaesthetic you used for each injection;”

- Provided a detailed explanation of the PECS block technique, including the dosage and type of anesthetic used, on Page 6, Line 20-22.

“At page 7, talking about muscle relaxation, include the neuromuscular monitoring;”

- Included a statement regarding the lack of neuromuscular monitoring on Page 7, Line 16-18.

‘At page 8, clarify what you did include in “adverse events”;’

- Clarified the definition of "adverse events" on Page 9, Line 1-2.

“In table n.1 you should also consider the QoR-15 pre-surgery;”

- Updated Table 1 to include pre-surgery QoR-15 scores.

“Consider a rearrangement of the Discussion paragraph as it’s too long compared to the overall length of the article.”

- Rearranged the Discussion section for improved clarity and conciseness on Page 18-20.

We believe these revisions adequately address the reviewers' concerns and enhance the quality of our manuscript. We appreciate the opportunity to improve our work and look forward to your feedback on our revised submission.

Sincerely,

Qingyan Luo

From the Department of Anesthesiology, Obstetrics and Gynecology Hospital of Fudan University, Shanghai, China.

Reference

1. Temple-Oberle C, Shea-Budgell M, Tan M, Semple J, Schrag C, Barreto M, et al. Consensus Review of Optimal Perioperative Care in Breast Reconstruction: Enhanced Recovery after Surgery (ERAS) Society Recommendations. Plastic and reconstructive surgery. 2017;139(5):1056e-71e.

2. Bu X, Zhang J, Zuo Y. Validation of the Chinese Version of the Quality of Recovery-15 Score and Its Comparison with the Post-Operative Quality Recovery Scale. The patient. 2016;9(3):251-9.

3. Qingfen Z, Yaqing W, Haiyan A, Yi F. Postoperative recovery after breast cancer surgery: A randomised controlled trial of opioid-based versus opioid-free anaesthesia with thoracic paravertebral block. Eur J Anaesthesiol. 2023;40(8).

---

## [Decision Letter · Decision Letter 1]

21 Feb 2025

Effect of opioid-free anesthesia combined with pectoral nerve block on the quality of recovery in patients after mastectomy: A randomized, controlled trial

PONE-D-24-44917R1

Dear Dr. Luo,

We’re pleased to inform you that your manuscript has been judged scientifically suitable for publication and will be formally accepted for publication once it meets all outstanding technical requirements.

Kind regards,

Alessandro De Cassai, MD

Academic Editor

PLOS ONE

Additional Editor Comments (optional):

According to Reviewers comments and my onw assessment your manuscript could be accepted

Reviewers' comments:

Reviewer's Responses to Questions

**Comments to the Author**

1. If the authors have adequately addressed your comments raised in a previous round of review and you feel that this manuscript is now acceptable for publication, you may indicate that here to bypass the “Comments to the Author” section, enter your conflict of interest statement in the “Confidential to Editor” section, and submit your "Accept" recommendation.

Reviewer #1: All comments have been addressed

Reviewer #2: All comments have been addressed

2. Is the manuscript technically sound, and do the data support the conclusions?

Reviewer #1: (No Response)

Reviewer #2: Yes

3. Has the statistical analysis been performed appropriately and rigorously? 

Reviewer #1: (No Response)

Reviewer #2: Yes

4. Have the authors made all data underlying the findings in their manuscript fully available?

Reviewer #1: (No Response)

Reviewer #2: Yes

5. Is the manuscript presented in an intelligible fashion and written in standard English?

Reviewer #1: (No Response)

Reviewer #2: Yes

6. Review Comments to the Author

Reviewer #1: (No Response)

Reviewer #2: Thank you for requesting this further revision. the authors have made the requested revisions. for me, the manuscript can be accepted.

7. PLOS authors have the option to publish the peer review history of their article (what does this mean? ). If published, this will include your full peer review and any attached files.

**Do you want your identity to be public for this peer review?** For information about this choice, including consent withdrawal, please see our Privacy Policy .

Reviewer #1: No

Reviewer #2: **Yes: ** Covotta Marco, MD, IRCCS Regina Elena National Cancer Institute, Rome, Italy

---

## [Editor Report · Acceptance letter]

PONE-D-24-44917R1

PLOS ONE

Dear Dr. Luo,

I'm pleased to inform you that your manuscript has been deemed suitable for publication in PLOS ONE. Congratulations! Your manuscript is now being handed over to our production team.

Kind regards,

on behalf of

Dr. Alessandro De Cassai

Academic Editor

PLOS ONE